# Framework for Greenhouse Gas Emissions Calculations in the Context of Road Freight Transport for the Automotive Industry

Jan Chocholac [1,*] , Roman Hruska [1], Stanislav Machalik [2], Dana Sommerauerova [1] and Petr Sohajek [3]

1. Department of Transport Management, Marketing and Logistics, Faculty of Transport Engineering, University of Pardubice, 532 10 Pardubice, Czech Republic; roman.hruska@upce.cz (R.H.); dana.sommerauerova@upce.cz (D.S.)
2. Department of Informatics and Mathematics in Transport, Faculty of Transport Engineering, University of Pardubice, 532 10 Pardubice, Czech Republic; stanislav.machalik@upce.cz
3. OLTIS Group a.s., 533 51 Pardubice-Rosice, Czech Republic; sohajek@oltis.cz
* Correspondence: jan.chocholac@upce.cz

**Abstract:** The topic of greenhouse gas emissions calculations in the context of freight transport is very current. This topic is very interesting for many stakeholders, such as companies, suppliers, employees, customers, residents, etc. The automotive industry is a major producer of greenhouse gas emissions from logistic processes. Due to this fact, it is necessary to search for and create frameworks for the calculation of greenhouse gas emissions in this sector. The requirements for the calculation of greenhouse gas emissions from road freight transport in the automotive industry were identified using semi-structured interviews. Available emission freight calculators were analyzed using the content and comparative analysis. The proposed frameworks for greenhouse gas emissions calculations in the context of road freight transport of material and finished manufactured passenger cars for the automotive industry were applied in the form of an interpretative case study. The main result of the article is the proposal of the frameworks for greenhouse gas (carbon and sulfur dioxide) emissions calculations in the context of road freight transport of the material and finished manufactured passenger cars for the automotive industry. The proposed frameworks were applied and verified. The use of the proposed frameworks can be expected in logistic planning and decision-making.

**Keywords:** logistics; transportation; road freight transport; greenhouse gas emissions; greenhouse gas emissions calculator; automotive industry

## 1. Introduction

As a result of global warming, increasing attention is being paid to one of its causes, and that is the constant increase in greenhouse gases (hereinafter GHG) due to various human activities [1]. The authors of the article focus on GHG emissions from road freight transport in the automotive industry with the aim to calculate them.

The transport sector is the second most important sector contributing to the production of $CO_2$ emissions worldwide [2]. It is responsible for more than 30% of the total energy consumption within the country members of the European Environment Agency [3].

According to the IEA (International Energy Agency) [4], global transport emissions increased by less than 0.5% in 2019 (compared with 1.9% annually since 2000) owing to efficiency improvements, electrification, and greater use of biofuels (Figure 1). Nevertheless, transport is still responsible for 24% of direct $CO_2$ emissions from fuel combustion [4]. In terms of transport modes, road vehicles account for nearly 75% of global transport emissions [5], and emissions from aviation and shipping continue to increase, but more emphasis on greater international policy focuses on these hard-to-abate subsectors [4].

The automotive industry has long been the most important sector of the economy in the Czech Republic, but also in the European Union. In the Czech Republic, it accounts for

approximately 10% of the gross domestic product (GDP) [6]. Many global and domestic suppliers are linked to the automotive industry. Suppliers involved in the production chain of the automotive industry of the Czech Republic participate 23% of the industrial production of the Czech Republic [6]. Large material flows (for example, components, finished cars) are associated with the automotive industry, which means a large amount of transport, and this creates GHG emissions.

Low emission logistics have become an expected and desired goal in all fields of transport, particularly in the European Union (EU) [7]. The EU is already on track to meet its GHG emissions reduction target for 2020 and has put forward a plan to further reduce emissions by at least 55% in 2030 [8].

Road transport is one of the main sources of air pollutants [9]. Emissions from road freight transport continue to rise [4]. This is illustrated in Figure 1.

Since 2019, the growth rate of emissions has continued to decline, with the forecast that emissions from transport will decrease in the near future. The emissions are also connected with freight volume and roadway infrastructure, which can be analyzed to assist the policymakers in reducing global emissions [10].

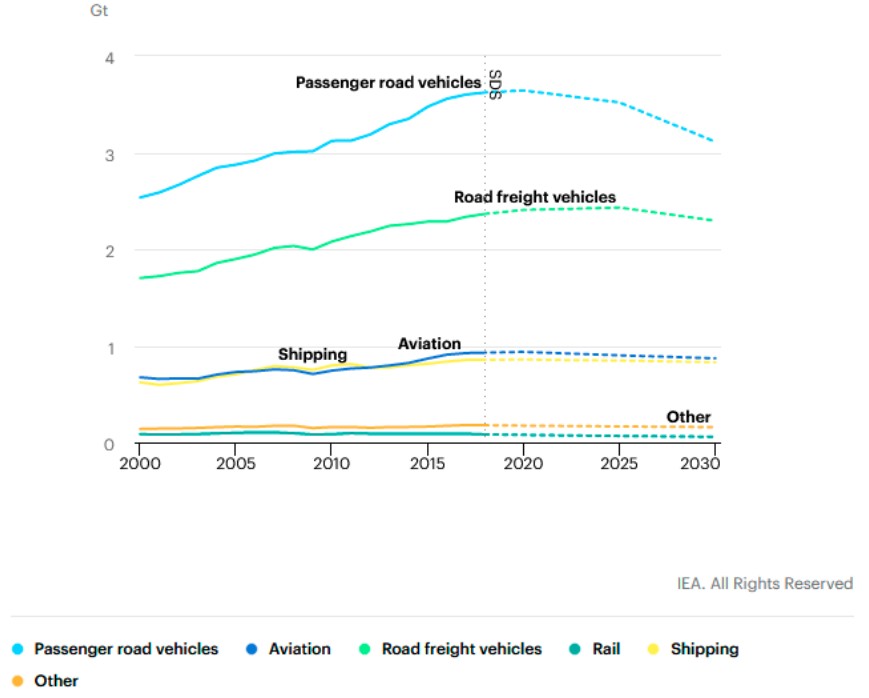

**Figure 1.** Transport sector $CO_2$ emissions by mode in the Sustainable Development Scenario, 2000–2030 [11].

Matthias et al. [12] focus on modelling road transport emissions in Germany until 2040 for the sustainable development of transport. Craglia and Cullen [13] deal with modeling transport emissions in Great Britain. Each country must ensure the sustainability of transport. For example, Poland must face the inevitable transition to renewable energy sources, which will decrease the consumption of fossil fuels and, consequently, GHG emissions from road transport [14].

European standard EN 16258 Methodology for the calculation and declaration of energy consumption and GHG emissions of transport services (freight and passengers) was approved by the European Committee for Standardization in 2012 [15,16]. Nowadays, there are three main approaches to measuring energy consumption and produced emissions—Well-to-Wheel (hereinafter also referred to as WtW), Well-to-Tank (hereinafter also referred to as WtT), and Tank-to-Wheel (hereinafter also referred to as TtW):

- Well-to-Wheel (total Well-to-Tank together with Tank-to-Wheel): An approach based on the monitoring of energy consumption and associated emissions production that covers the whole process from the generation of electricity or fuel, through the supply to the appropriate transport means through the distribution network, to the consumption associated with the operation of the means of transport. This approach is based on the sum of Tank-to-Wheel and Well-to-Tank values (Figure 2) [16].
- Well-to-Tank: Energy consumption and production of emissions related to the production of energy or fuel—the indicator covers all activities from the extraction of raw materials through the production of energy or fuel, up to the supply to the respective means of transport through the distribution network. The indicator does not include the transport mode (Figure 2) [16].
- Tank-to-Wheel: Energy consumption and production of emissions related to the operation of the means of transport. The indicator does not include the next life cycle of the fuel or transport means (Figure 2) [16].

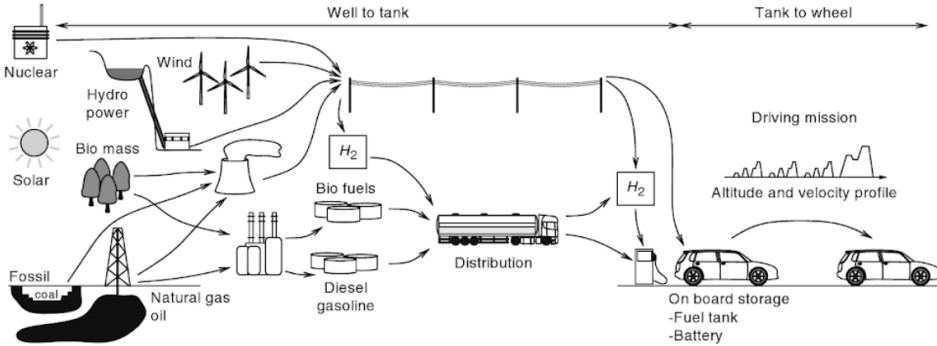

**Figure 2.** Well-to-Tank and Tank-to-Wheel [16].

The framework for GHG emissions calculations in the context of road freight transport should identify the relationship between the financial aspect and the level of generated GHG emissions [17]. Important user inputs for GHG emissions calculation are the type of vehicle, emission standard, load weight and transport distance. Longer transport distances lead to increased vehicle emissions on transportation routes, resulting in a greater carbon footprint [18].

A reduction in GHG emissions from freight transport can be achieved by pooling supply chains [19].

Calculators of GHG emissions are important tools for estimating GHG emissions and for providing information that can lead to behavioral and policy change [20,21].

The aim of this article is to propose a framework for greenhouse gas emissions calculations in the context of road freight transport for the automotive industry.

## 2. Materials and Methods

The processing procedure consists of five steps (Figure 3). The requirements for the calculation of GHG emissions from road freight transport in the automotive industry are identified in the first step using the scientific method of semi-structured interviews (theoretically described in Section 2.1). Available emission freight calculators are analyzed in the second step using the scientific method of content and comparative analysis (theoretically described in Sections 2.2 and 2.3). The frameworks for GHG emissions calculations in the context of road freight transport of the material and finished manufactured passenger cars (hereinafter FMPC) for the automotive industry are proposed in the third step. The frameworks for carbon and sulfur dioxide emissions calculations in the context of road freight transport for the automotive industry are proposed in the fourth step.

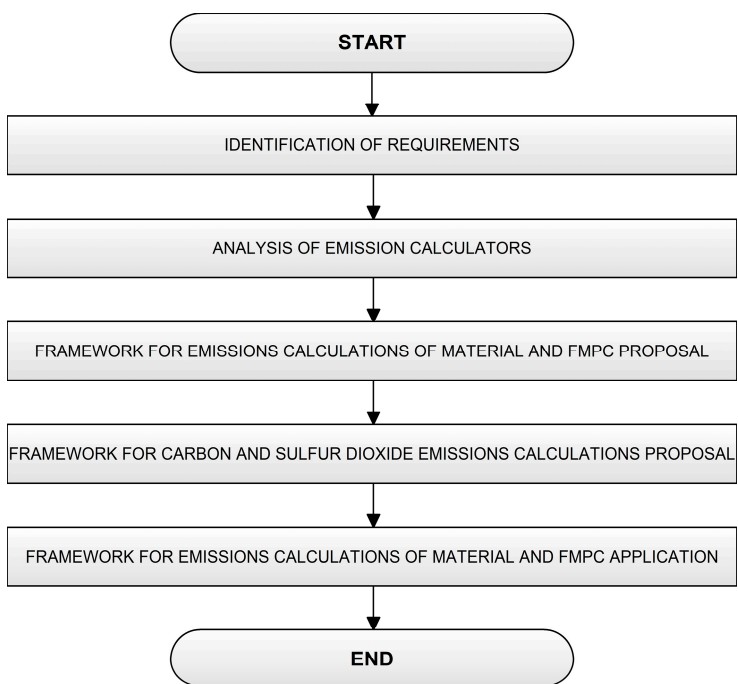

**Figure 3.** Processing procedure [authors].

The frameworks for GHG emissions calculations in the context of road freight transport of the material and FMPC for the automotive industry are applied in the fifth step using the scientific method of interpretative case study (theoretically described in Section 2.4). The processing procedure is shown in Figure 3.

The scientific methods used in this research are theoretically defined in the following sections.

## 2.1. Semi-Structured Interviews

The scientific method of semi-structured interviews is used to identify requirements for the calculation of GHG emissions from road freight transport with respondents from a leading company in the automotive industry operating on the market in the Czech Republic.

The method of semi-structured interviews is highly recommended in the situation where researchers were able to narrow down some areas or topics of the research because a completely unstructured interview has the risk of not eliciting the topics or themes more closely related to the research questions under consideration [22]. The sample size in qualitative studies may vary, but within semi-structured interviews, a small sample is sufficient [23].

The method of semi-structured interviews was, for example, used by Naghavi et al. [24] for data collection from pilots and safety personnel, Aljohani and Thompson [25] for the examination of last mile delivery practices of freight carriers servicing business receivers in inner-city areas, Nakanishi and Black [26] for travel habit creation of the elderly and the transition to sustainable transport, and Hilgarter and Granig [27] to obtain the public perception of autonomous vehicles after riding an autonomous shuttle.

## 2.2. Content Analysis

The scientific method of content analysis is used to analyze available emission freight calculators. The method of content analysis is based on the depth examination of sources [28]. This allows the collection and analysis of whatever user-generated content as electronic information [29]. The abundance of text available in the online space builds the basis for content analysis [30]. Content analysis is one of the most commonly used methods to deal with qualitative data [31].

This method was, for example, used by Leung et al. for the research processing in the area of fuel price changes and their impacts on urban transport, Zhou and Wang to explore the logistic market, Gao et al. to explore the public participation in smart-city governance in Urban China, and Fu and Zhang to analyze master plans of eco, low-carbon and conventional new towns in China [28,32–34].

### 2.3. Comparative Analysis

The scientific method of comparative analysis is used to analyze and compare available emission freight calculators. The method of comparative analysis is a data analysis technique for determining which logical conclusions a data set supports [35]. The method begins with listing all the combinations of variables observed in the data set, followed by applying the rules of logical inference to determine which descriptive inferences or implications the data support [35].

The method of comparative analysis was, for example, used by Carvalho and Medeiros to evaluate two models (SERVQUAL and SERVPERF) to investigate the factors that influence the formation of perceived quality in airline services, Shah et al. to compare and analyze both conventional fuel and hybrid bus systems for the Multan city of Pakistan, Zitricky et al. to assess railway and air passenger transport operation in terms of environmental impact, and Mintzia et al. to investigate the $CO_2$ impact of the construction and operation of the main highway and railway line infrastructure in Greece [36–39].

### 2.4. Interpretative Case Study

The scientific method of interpretative case study is used to demonstrate and verify the proposed framework for GHG emissions (carbon and sulfur dioxide) calculations in the context of road freight transport of the material and FMPC for the automotive industry.

Using interpretive case studies is therefore perfectly justified for exploratory research [40]. The emerging nature of research in companies is best suited to an interpretative qualitative approach that can yield a rich understanding of key issues by minimizing the distance between the researcher and the key decision-maker, the owner/manager, in order to develop the practical and theoretical understanding and generate new and alternative theories and concepts [41]. The approach can be used regardless of the size of the company. When using interpretative case studies for exploratory research, key decisions for the researcher concern the role of prior theory, the unit(s) of analysis, the number and selection of cases, the techniques to be used for data collection, and the method(s) by which the collected data will be analyzed [42].

The method of interpretative case study was, for example, used by Melo et al. for investigation factors that affect agile team productivity in three large companies in Brazil [43].

## 3. Results

The Results chapter consists of these sections: identification of requirements for the calculation of greenhouse gas emissions from road freight transport (Section 3.1), analysis of available emission freight calculators (Section 3.2), framework for greenhouse gas emissions calculations in the context of road freight transport of material and finished manufactured passenger cars for the automotive industry proposal (Sections 3.3 and 3.4), framework for carbon dioxide and sulfur dioxide emissions calculations in the context of road freight transport for the automotive industry proposal (Sections 3.5 and 3.6), framework for greenhouse gas emissions calculations in the context of road freight transport of material and finished manufactured passenger cars for the automotive industry application (Sections 3.7 and 3.8).

### 3.1. Identification of Requirements for the Calculation of Greenhouse Gas Emissions from Road Freight Transport

Two independent researchers used the two-round semi-structured interview in October 2020 with three respondents from a leading company in the automotive industry operating on the market in the Czech Republic [44–46].

The semi-structured interview covered the following areas:

- Calculating GHG emissions from road transport;
- Using GHG emissions calculators;
- Cargo types;
- Vehicle types and their specifications;
- Transport restrictive conditions and limitations;
- GHG emissions calculation requirements.

The specific questions used in the semi-structured interview are in Appendix A. The aggregated answers are shown in Appendix B. An overview of the specifications and parameters of the vehicles used for road freight transport is shown in Table 1. The overview is divided into vehicles for the transport of FMPC and material.

**Table 1.** An overview of the specifications and parameters of the vehicles used for road freight transport [authors].

| Type of Cargo | Vehicle | Maximum Load Weight | Maximum Load Volume |
|---|---|---|---|
| Finished manufactured passenger cars (FMPC) | $V_1$ | 20,000 kg | 8 or 9 cars by type |
| | $V_2$ | 11,500 kg | 4 cars |
| | $V_3$ | 8000 kg | 2 cars |
| Material | $V_4$ | 24,000 kg | 150 m$^3$ |
| | $V_5$ | 24,000 kg | 100 m$^3$ |
| | $V_6$ | 24,000 kg | 120 m$^3$ |
| | $V_7$ | 24,000 kg | 80 m$^3$ |
| | $V_8$ | 12,000 kg | 120 m$^3$ |
| | $V_9$ | 3350 kg | 38 m$^3$ |

The most important conclusions obtained from the semi-structured interview are as follows: this issue is very current for the leading company in the automotive industry operating on the market in the Czech Republic. The company does not currently use any road transport emissions freight calculators because there is no suitable emission freight calculator available to meet the company's requirements. The logistic processes of the company are very extensive and specific with many conditions. Currently, there is no freight emission calculator that contains all specifics.

### 3.2. Analysis of Available Emission Freight Calculators

Three independent researchers used the content analysis and comparative analysis of emission freight calculators available in December 2020 based on the outputs of the semi-structured interview. These emission freight calculators were considered:

- No. 1—$CO_2$ emission calculator—EECA Business [47];
- No. 2—Emission calculator and carbon offset—SAS [48];
- No. 3—Emissions Calculator—Cargolux [49];
- No. 4—Business $CO_2$ emissions calculator—ClimateCare [50];
- No. 5—Freight Emissions Calculator [51];
- No. 6—Emission Calculators—Sustainable Freight [52];
- No. 7—EcoTransIT World—Calculation [53];
- No. 8—Van Donge & De Roo calculator [54];
- No. 9—OOCL Carbon Calculator [55];
- No. 10—Carbon Calculator—cn.ca [56].

The results of the content analysis of emission freight calculators are presented in Table 2. If the emission freight calculator did not allow the calculation of emissions from road freight transport (emission freight calculator no. 2, 3, 6, 8 and 9), it was not further analyzed, respectively, not part of Table 2. Monitored parameters, which are important for the research, are shown in the columns.

**Table 2.** The results of the content analysis of emission freight calculators (authors based on [47–56]).

| No. | Source | Road Transport | Transport of FMPC and Material | Own Vehicle | Implementation of Restrictive Conditions | One-Way and Round Trip Transport | $CO_{2e}$ and $SO_{2e}$ Outputs | Total and Average Emissions Outputs | WtW, WtT and TtW Calculation Approach | Monetization of Produced Emissions |
|---|---|---|---|---|---|---|---|---|---|---|
| 1 | [47] | Y | NA | NA | NA | Only one-way | Only $CO_2$ | Only total emissions | NA | NA |
| 4 | [50] | Y | NA | NA | NA | Only one-way | Only $CO_2$ | Only total emissions | NA | Y |
| 5 | [51] | Y | NA | NA | NA | Only one-way | Only $CO_2$ | Only total emissions | NA | NA |
| 7 | [53] | Y | NA | Y | NA | Y | Y | Only total emissions | Y | NA |
| 10 | [56] | Y | NA | NA | NA | Only one-way | Only $CO_2$ | Only total emissions | NA | NA |

Notes: FMPC—finished manufactured passenger cars, $CO_{2e}$—carbon dioxide equivalent, $SO_{2e}$—sulfur dioxide equivalent, WtW—Well-to-Wheel approach, WtT—Well-to-Tank approach, TtW—Tank-to-Wheel approach, Y—Yes, N—No, NA—not available, NFA—not further analyzed due to inapplicability to road transport.

None of the analyzed emission freight calculators allow the transport of FMPC or material and none of the analyzed emission freight calculators allow the implementation of restrictive conditions according to the required specification. Only one emission freight calculator (no. 7) allows you to enter the vehicle parameters according to your specification, calculate emissions for one-way and round trip transport, calculate the emissions produced as $CO_{2e}$ and $SO_{2e}$ outputs and use three different calculation approaches; but it does not meet other important requirements. Only one emission freight calculator (no. 4) allows you to monetize produced emissions, but it does not meet other important requirements.

In conclusion, it can be stated that none of the analyzed emission freight calculators are suitable for use in companies in the automotive industry because they do not meet all the conditions and requirements (specified in Table 2). This statement was also confirmed by the results of semi-structured interviews (Appendices A and B). Based on these facts, a market gap was identified.

*3.3. Framework for Greenhouse Gas Emissions Calculations in the Context of Road Freight Transport of the Material for the Automotive Industry Proposal*

The framework for GHG emissions calculations in the context of road freight transport for the automotive industry proposal is presented in Figure 4. The calculation of GHG emissions from road freight transport starts with the selection of the vehicle (the first step in Figure 4). The vehicle is selected by the user from a predefined menu. In the second step, the user selects the emission standard of the vehicle from a predefined menu of emission standards. Based on the selected vehicle and the emission standard, related parameters and specifications are retrieved from the database. In the third step, the user enters either the weight of the transported material or the number of transported FMPC according to their types. Based on the entered inputs, predefined restrictive conditions and limitations are checked. If they are not met, it is necessary to modify the entered inputs. If they are met, the next step proceeds. In the fourth and fifth step, the transport distance and the type of transport are entered (whether it is a round trip or one-way transport). Furthermore, the vehicle load factor is also calculated and the emission coefficients are searched in the database. Finally, emissions are calculated and the results are presented. Further in the text, the restrictive conditions and limitations and the method of searching for emission coefficients in the database are given.

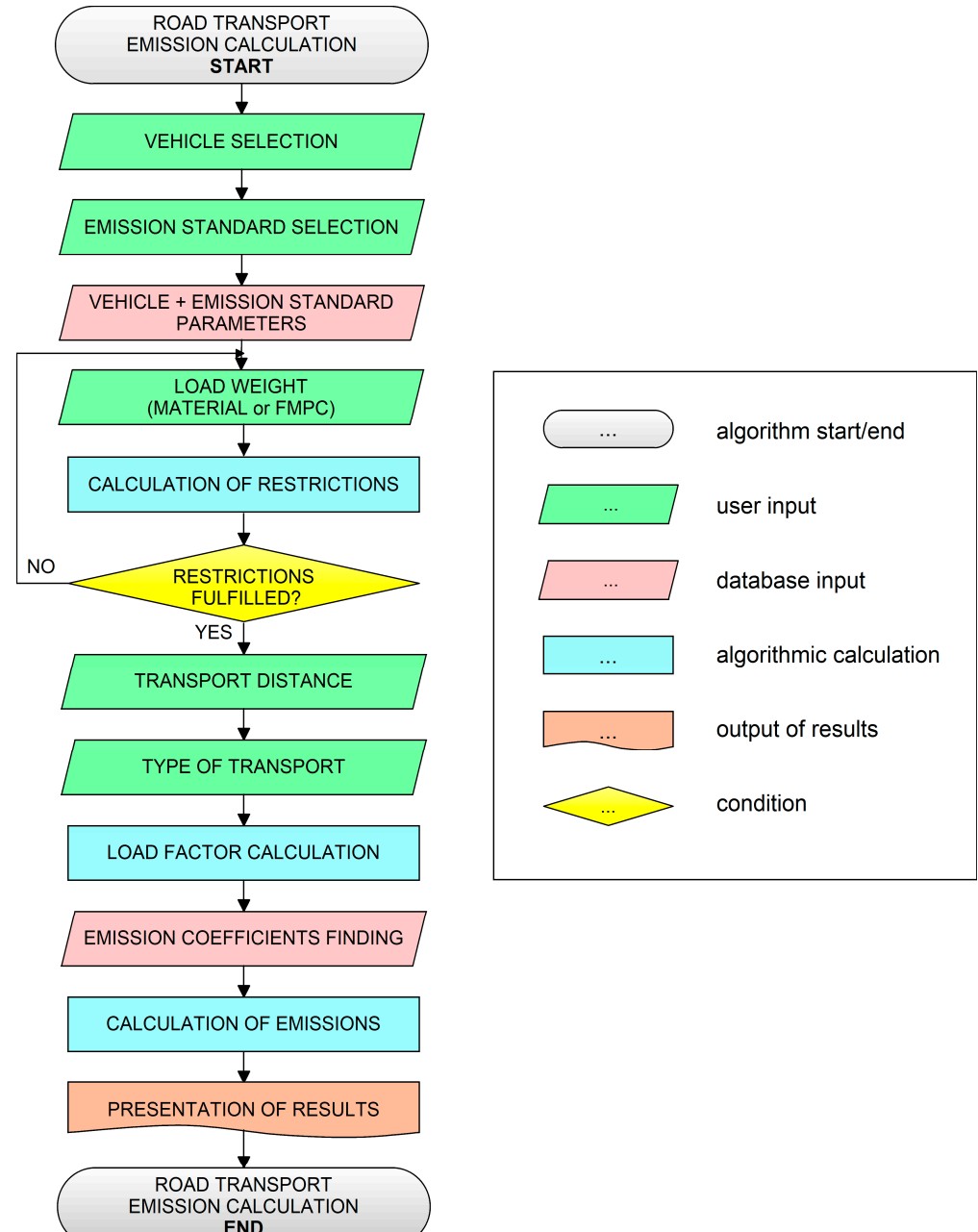

**Figure 4.** Framework for greenhouse gas emissions calculations in the context of road freight transport for the automotive industry [authors].

Concerning the transported weight of the material, the following restrictive conditions and limitations must be respected:

$$
\begin{aligned}
\text{for } V_4 \text{ to } V_7: W \leq W_{max} \leq 24{,}000 \text{ [kg]}, \\
\text{for } V_8: W \leq W_{max} \leq 12{,}000 \text{ [kg]}, \\
\text{for } V_9: W \leq W_{max} \leq 3350 \text{ [kg]},
\end{aligned}
\tag{1}
$$

where $V_{4-9}$ represent vehicles for the transportation of material (the specification and parameters of the vehicles are shown in Table 1), W is the weight of the material and $W_{max}$ is the maximum load weight of the vehicle.

Concerning the transported volume of material, the following restrictive conditions and limitations must be respected:

$$
\begin{aligned}
&\text{for } V_4: \text{LV} \leq \text{LV}_{max} \leq 150 \ [\text{m}^3], \\
&\text{for } V_5: \text{LV} \leq \text{LV}_{max} \leq 100 \ [\text{m}^3], \\
&\text{for } V_6: \text{LV} \leq \text{LV}_{max} \leq 120 \ [\text{m}^3], \\
&\text{for } V_7: \text{LV} \leq \text{LV}_{max} \leq 80 \ [\text{m}^3], \\
&\text{for } V_8: \text{LV} \leq \text{LV}_{max} \leq 120 \ [\text{m}^3], \\
&\text{for } V_9: \text{LV} \leq \text{LV}_{max} \leq 38 \ [\text{m}^3],
\end{aligned}
\tag{2}
$$

where $V_{4-9}$ represent vehicles for the transportation of material (the specification and parameters of the vehicles are shown in Table 1), LV is the load volume of the material and $\text{LV}_{max}$ is the maximum load volume of the vehicle.

The vehicle load factor is calculated as follows:

$$
\text{LF}_i = \text{round up } [W/W_{max}] \ [-], \ \text{LF}_i \in \ <0;1>, \ i = \ <1; \infty),
\tag{3}
$$

where $\text{LF}_i$ is the vehicle load factor of vehicle i, i is the type of the vehicle, W is the weight of the material and $W_{max}$ is the maximum load weight of the vehicle.

The emission coefficients are searched in the database according to the following algorithmic procedures from the perspective of the maximum load weight of the vehicle:

$$
\begin{aligned}
&\text{IF vehicle } V_4 \text{ to } V_7 \text{ THEN emission coefficients for 24 t vehicles,} \\
&\text{IF vehicle } V_8 \text{ THEN emission coefficients for 12 t vehicles,} \\
&\text{IF vehicle } V_9 \text{ THEN emission coefficients for 3.35 t vehicles,}
\end{aligned}
\tag{4}
$$

where $V_{4-9}$ represent vehicles for the transportation of the material.

The emission coefficients are searched in the database according to the following algorithmic procedures from the perspective of the emission standard of the vehicle:

$$
\begin{aligned}
&\text{IF emission standard "EURO 3" THEN emission coefficients for "EURO 3",} \\
&\text{IF emission standard "EURO 4" THEN emission coefficients for "EURO 4",} \\
&\text{IF emission standard "EURO 5" THEN emission coefficients for "EURO 5",} \\
&\text{IF emission standard "EURO 6" THEN emission coefficients for "EURO 6",}
\end{aligned}
\tag{5}
$$

where EURO 3–6 are emission standards.

The emission coefficients are searched in the database according to the following algorithmic procedures (Equation (6)) after the aggregation of conditions (Equations (4) and (5)):

$$
\begin{aligned}
&\text{IF vehicle } V_4 \text{ to } V_7 \text{ AND emission standard "EURO 3" THEN emission} \\
&\qquad\qquad \text{coefficients for "EURO 3" 24 t vehicles,} \\
&\text{IF vehicle } V_4 \text{ to } V_7 \text{ AND emission standard "EURO 4" THEN emission} \\
&\qquad\qquad \text{coefficients for "EURO 4" 24 t vehicles,} \\
&\qquad\qquad\qquad\qquad\qquad \vdots \\
&\text{IF vehicle } V_4 \text{ to } V_7 \text{ AND emission standard "EURO 6" THEN emission} \\
&\qquad\qquad \text{coefficients for "EURO 6" 24 t vehicles,}
\end{aligned}
\tag{6}
$$

where $V_{4-7}$ represent vehicles for the transportation of material.

The emission coefficients are searched in the database according to the following algorithmic procedures (Equation (7)) after the aggregation of conditions (Equations (4) and (5)):

$$
\begin{aligned}
&\text{IF vehicle } V_8 \text{ AND emission standard "EURO 3" THEN emission} \\
&\qquad\qquad \text{coefficients for "EURO 3" 12 t vehicles,} \\
&\text{IF vehicle } V_8 \text{ AND emission standard "EURO 4" THEN emission} \\
&\qquad\qquad \text{coefficients for "EURO 4" 12 t vehicles,} \\
&\qquad\qquad\qquad\qquad\qquad \vdots \\
&\text{IF vehicle } V_8 \text{ AND emission standard "EURO 6" THEN emission} \\
&\qquad\qquad \text{coefficients for "EURO 6" 12 t vehicles,}
\end{aligned}
\tag{7}
$$

where $V_8$ is the vehicle for the transportation of material.

The emission coefficients are searched in the database according to the following algorithmic procedures (Equation (8)) after the aggregation of conditions (Equations (4) and (5)):

$$
\begin{aligned}
&\text{IF vehicle } V_9 \text{ AND emission standard "EURO 3" THEN emission} \\
&\qquad\text{coefficients for "EURO 3" 3.35 t vehicles,} \\
&\text{IF vehicle } V_9 \text{ AND emission standard "EURO 4" THEN emission} \\
&\qquad\text{coefficients for "EURO 4" 3.35 t vehicles,} \\
&\qquad\qquad\qquad\qquad\vdots \\
&\text{IF vehicle } V_9 \text{ AND emission standard "EURO 6" THEN emission} \\
&\qquad\text{coefficients for "EURO 6" 3.35 t vehicles,}
\end{aligned}
\tag{8}
$$

where $V_9$ is the vehicle for the transportation of material.

The emission coefficients are further searched in the database according to the following algorithmic procedures concerning the type of transport:

$$
\begin{aligned}
&\text{IF one-way transport THEN find values of emission coefficients} \\
&\text{for } EC_{WtTb}, EC_{WtTf}, EC_{TtWb}, EC_{TtWf}, EC_{0WtTb}, EC_{0WtTf}, EC_{0TtWb} \text{ and } EC_{0TtWf}, \\
&\text{IF round trip transport THEN find values of emission coefficients} \\
&\text{for } EC_{WtTb}, EC_{WtTf}, EC_{TtWb} \text{ and } EC_{TtWf},
\end{aligned}
\tag{9}
$$

where $EC_{WtTb}$ is the relevant emission coefficient of the biogenic origin calculated using the WtT approach, $EC_{WtTf}$ is the relevant emission coefficient of the fossil origin calculated using the WtT approach, $EC_{TtWb}$ is the relevant emission coefficient of the biogenic origin calculated using the TtW approach, $EC_{TtWf}$ is the relevant emission coefficient of the fossil origin calculated using the TtW approach, $EC_{0WtTb}$ is the relevant emission coefficient for the zero load of the biogenic origin calculated using the WtT approach, $EC_{0WtTf}$ is the relevant emission coefficient for the zero load of the fossil origin calculated using the WtT approach, $EC_{0TtWb}$ is the relevant emission coefficient for the zero load of the biogenic origin calculated using the TtW approach, and $EC_{0TtWf}$ is the relevant emission coefficient for the zero load of the fossil origin calculated using the TtW approach.

The application of the framework for GHG emissions calculations in the context of road freight transport of the material for the automotive industry is presented in Section 3.7.

### 3.4. Framework for Greenhouse Gas Emissions Calculations in the Context of Road Freight Transport of Finished Manufactured Passenger Cars for the Automotive Industry Proposal

The framework for GHG emissions calculations in the context of road freight transport for the automotive industry proposal is presented in Figure 4. Concerning the transported weight of FMPC, the following restrictive conditions and limitations must be respected:

$$
\begin{aligned}
&\text{for } V_1: W \leq W_{max} \leq 20{,}000 \text{ [kg],} \\
&\text{for } V_2: W \leq W_{max} \leq 11{,}500 \text{ [kg],} \\
&\text{for } V_3: W \leq W_{max} \leq 8000 \text{ [kg],}
\end{aligned}
\tag{10}
$$

where $V_{1–3}$ represent vehicles for the transportation of FMPC (the specification and parameters of the vehicles are shown in Table 1), W is the weight of the FMPC and $W_{max}$ is the maximum load weight of the vehicle.

Concerning the transported volume of FMPC, the following restrictive conditions and limitations must be respected:

$$
\begin{aligned}
&\text{IF } V_1 \text{ AND } (PC_1 + PC_2) < 3 \text{ THEN for } V_1: LV \leq LV_{max} \leq 8 \text{ [−],} \\
&\text{IF } V_1 \text{ AND } (PC_1 + PC_2) \geq 3 \text{ THEN for } V_1: LV \leq LV_{max} \leq 9 \text{ [−],} \\
&\qquad\qquad\text{IF } V_2 \text{ THEN } LV \leq LV_{max} \leq 4 \text{ [−],} \\
&\qquad\qquad\text{IF } V_3 \text{ THEN } LV \leq LV_{max} \leq 2 \text{ [−],}
\end{aligned}
\tag{11}
$$

where $V_{1-3}$ represent vehicles for the transportation of FMPC (the specification and parameters of the vehicles are shown in Table 1), LV is the load volume of the FMPC and $LV_{max}$ is the maximum load volume of the vehicle.

The required number of vehicles LKW for FMPC transport is calculated as follows:

$$
\begin{aligned}
&\text{IF}_1 \ V_2 \ \text{THEN round up } [\textstyle\sum PC_j/LV_{max}] = LKW \ [-], \ j = <1; \infty), \ LKW \in N, \\
&\text{IF}_1 \ V_3 \ \text{THEN round up } [\textstyle\sum PC_j/LV_{max}] = LKW \ [-], \ j = <1; \infty), \ LKW \in N, \\
&\text{IF}_1 \ V_1 \ \text{AND } (PC_1 + PC_2) < 3 \ \text{THEN for } V_1: \text{round up } [\textstyle\sum PC_j/LV_{max}] = LKW \ [-], \ j = <1; \infty), \ LKW \in N, \\
&\text{IF}_1 \ V_1 \ \text{AND } (PC_1 + PC_2) \geq 3 \ \text{THEN round down } [(PC_1 + PC_2)/3] = LKW_9, \ LKW_9 \cdot 9 = PLKW_9 \ [-], \\
&\text{IF}_2 \ PLKW_9 = \textstyle\sum PC_j \ \text{THEN } LKW_9 = LKW, \ LKW \in N, \\
&\text{IF}_2 \ PLKW_9 < \textstyle\sum PC_j \ \text{THEN } [\text{round up } (\textstyle\sum PC_j - PLKW_9)/8] + LKW_9 = LKW, \ LKW \in N, \\
&\text{IF}_2 \ PLKW_9 > \textstyle\sum PC_j \ \text{AND IF}_3 \ PLKW_9 - \textstyle\sum PC_j \geq 9 \ \text{THEN} \\
&LKW_9 - [\{\text{round down } (PLKW_9 - \textstyle\sum PC_j)\}/9] = LKW, \ LKW \in N, \\
&\text{IF}_2 \ PLKW_9 > \textstyle\sum PC_j \ \text{AND IF}_3 \ PLKW_9 - \textstyle\sum PC_j < 9 \ \text{THEN } LKW_9 = LKW, \\
&LKW \in N,
\end{aligned}
\tag{12}
$$

where $V_{1-3}$ represent vehicles for the transportation of FMPC (the specification and parameters of the vehicles are shown in Table 1), $PC_j$ is the number of FMPC type j, j is the type of transported FMPC, $LV_{max}$ is the maximum load volume of the vehicle, LKW is the number of vehicles for FMPC transport, $LKW_9$ is the number of vehicles for nine FMPC transport and $PLKW_9$ is the number of places in the $LKW_9$ vehicle for FMPC transport.

Then, the average vehicle load factor is calculated as follows:

$$
LF_i = \text{round up } [\textstyle\sum W/\textstyle\sum W_{max}] \ [-], \ LF_i \in <0;1>, \ i = <1; \infty),
\tag{13}
$$

where $LF_i$ is the average vehicle load factor of vehicle i, i is the type of vehicle, W is the weight of the FMPC and $W_{max}$ is the maximum load weight of the vehicle.

The emission coefficients are searched in the database according to the following algorithmic procedures from the perspective of the maximum load weight of the vehicle:

$$
\begin{aligned}
&\text{IF vehicle } V_1 \ \text{THEN emission coefficients for 20 t vehicles,} \\
&\text{IF vehicle } V_2 \ \text{THEN emission coefficients for 11.5 t vehicles,} \\
&\text{IF vehicle } V_3 \ \text{THEN emission coefficients for 8 t vehicles,}
\end{aligned}
\tag{14}
$$

where $V_{1-3}$ represent vehicles for the transportation of FMPC.

The emission coefficients are searched in the database according to the following algorithmic procedures from the perspective of the emission standard of the vehicle:

$$
\begin{aligned}
&\text{IF emission standard "EURO 3" THEN emission coefficients for "EURO 3",} \\
&\text{IF emission standard "EURO 4" THEN emission coefficients for "EURO 4",} \\
&\text{IF emission standard "EURO 5" THEN emission coefficients for "EURO 5",} \\
&\text{IF emission standard "EURO 6" THEN emission coefficients for "EURO 6",}
\end{aligned}
\tag{15}
$$

where EURO 3–6 are emission standards.

The emission coefficients are searched in the database according to the following algorithmic procedures (Equation (16)) after the aggregation of conditions (Equations (14) and (15)):

$$
\begin{aligned}
&\text{IF vehicle } V_1 \ \text{AND emission standard "EURO 3" THEN emission} \\
&\qquad \text{coefficients for "EURO 3" 20 t vehicles,} \\
&\text{IF vehicle } V_1 \ \text{AND emission standard "EURO 4" THEN emission} \\
&\qquad \text{coefficients for "EURO 4" 20 t vehicles,} \\
&\qquad\qquad\qquad\qquad : \\
&\text{IF vehicle } V_1 \ \text{AND emission standard "EURO 6" THEN emission} \\
&\qquad \text{coefficients for "EURO 6" 20 t vehicles,}
\end{aligned}
\tag{16}
$$

where $V_1$ is the vehicle for the transportation of FMPC.

The emission coefficients are searched in the database according to the following algorithmic procedures (Equation (17)) after the aggregation of conditions (Equations (14) and (15)):

$$
\begin{aligned}
&\text{IF vehicle } V_2 \text{ AND emission standard "EURO 3" THEN emission} \\
&\qquad \text{coefficients for "EURO 3" 11.5 t vehicles,} \\
&\text{IF vehicle } V_2 \text{ AND emission standard "EURO 4" THEN emission} \\
&\qquad \text{coefficients for "EURO 4" 11.5 t vehicles,} \\
&\qquad\qquad\qquad\qquad\qquad\qquad \vdots \\
&\text{IF vehicle } V_2 \text{ AND emission standard "EURO 6" THEN emission} \\
&\qquad \text{coefficients for "EURO 6" 11.5 t vehicles,}
\end{aligned}
\tag{17}
$$

where $V_2$ is the vehicle for the transportation of FMPC.

The emission coefficients are searched in the database according to the following algorithmic procedures (Equation (18)) after the aggregation of conditions (Equations (14) and (15)):

$$
\begin{aligned}
&\text{IF vehicle } V_3 \text{ AND emission standard "EURO 3" THEN emission} \\
&\qquad \text{coefficients for "EURO 3" 8 t vehicles,} \\
&\text{IF vehicle } V_3 \text{ AND emission standard "EURO 4" THEN emission} \\
&\qquad \text{coefficients for "EURO 4" 8 t vehicles,} \\
&\qquad\qquad\qquad\qquad\qquad\qquad \vdots \\
&\text{IF vehicle } V_3 \text{ AND emission standard "EURO 6" THEN emission} \\
&\qquad \text{coefficients for "EURO 6" 8 t vehicles,}
\end{aligned}
\tag{18}
$$

where $V_3$ is the vehicle for the transportation of FMPC.

The emission coefficients are further searched in the database according to the following algorithmic procedures concerning the type of transport:

$$
\begin{aligned}
&\text{IF one-way transport THEN find values of emission coefficients} \\
&\text{for } EC_{WtTb}, EC_{WtTf}, EC_{TtWb}, EC_{TtWf}, EC_{0WtTb}, EC_{0WtTf}, EC_{0TtWb} \text{ and } EC_{0TtWf}, \\
&\text{IF round trip transport THEN find values of emission coefficients} \\
&\text{for } EC_{WtTb}, EC_{WtTf}, EC_{TtWb} \text{ and } EC_{TtWf},
\end{aligned}
\tag{19}
$$

where $EC_{WtTb}$ is the relevant emission coefficient of the biogenic origin calculated using the WtT approach, $EC_{WtTf}$ is the relevant emission coefficient of the fossil origin calculated using the WtT approach, $EC_{TtWb}$ is the relevant emission coefficient of the biogenic origin calculated using the TtW approach, $EC_{TtWf}$ is the relevant emission coefficient of the fossil origin calculated using the TtW approach, $EC_{0WtTb}$ is the relevant emission coefficient for the zero load of the biogenic origin calculated using the WtT approach, $EC_{0WtTf}$ is the relevant emission coefficient for the zero load of the fossil origin calculated using the WtT approach, $EC_{0TtWb}$ is the relevant emission coefficient for the zero load of the biogenic origin calculated using the TtW approach, and $EC_{0TtWf}$ is the relevant emission coefficient for the zero load of the fossil origin calculated using the TtW approach.

The application of the framework for GHG emissions calculations in the context of road freight transport of FMPC for the automotive industry is presented in Section 3.8.

### 3.5. Framework for Carbon Dioxide Emissions Calculations in the Context of Road Freight Transport for the Automotive Industry Proposal

Carbon dioxide emissions are calculated for a specific shipment. In the case of round trip transportation, Equations (20)–(27) are used. In the case of one-way transportation, Equations (28)–(36) are used.

The total carbon dioxide emissions produced by round trip transportation are calculated as:

$$
E = E_1 \ [kg_{CO2e}],
\tag{20}
$$

where E represents the total carbon dioxide emissions produced by transportation and $E_1$ are the total carbon dioxide emissions produced by the round trip transportation.

$E_1$ are calculated using the WtW approach:

$$E_1 = E_{1WtT} + E_{1TtW} \ [kg_{CO2e}], \tag{21}$$

where $E_{1WtT}$ are the carbon dioxide emissions calculated using the WtT approach and $E_{1TtW}$ are the carbon dioxide emissions calculated using the TtW approach.

$E_{1WtT}$ are the carbon dioxide emissions calculated using the WtT approach and consisting of two components:

$$E_{1WtT} = E_{1WtTb} + E_{1WtTf} \ [kg_{CO2e}], \tag{22}$$

where $E_{1WtTb}$ are the carbon dioxide emissions of biogenic origin calculated using the WtT approach and $E_{1WtTf}$ are the carbon dioxide emissions of fossil origin calculated using the WtT approach.

$E_{1TtW}$ are the carbon dioxide emissions calculated using the TtW approach and consisting of two components:

$$E_{1TtW} = E_{1TtWb} + E_{1TtWf} \ [kg_{CO2e}], \tag{23}$$

where $E_{1TtWb}$ are the carbon dioxide emissions of biogenic origin calculated using the TtW approach and $E_{1TtWf}$ are the carbon dioxide emissions of fossil origin calculated using the TtW approach.

$E_{1WtTb}$ are calculated as follows:

$$E_{1WtTb} \ [kg_{CO2e}] = EC_{WtTb} \ [kg_{CO2e}/tkm] \cdot W \ [t] \cdot D_1 \ [km], \tag{24}$$

where $EC_{WtTb}$ is the relevant emission coefficient, W is the weight of the load and $D_1$ is the transport distance.

$E_{1WtTf}$ are calculated as follows:

$$E_{1WtTf} \ [kg_{CO2e}] = EC_{WtTf} \ [kg_{CO2e}/tkm] \cdot W \ [t] \cdot D_1 \ [km], \tag{25}$$

where $EC_{WtTf}$ is the relevant emission coefficient, W is the weight of the load and $D_1$ is the transport distance.

$E_{1TtWb}$ are calculated as follows:

$$E_{1TtWb} \ [kg_{CO2e}] = EC_{TtWb} \ [kg_{CO2e}/tkm] \cdot W \ [t] \cdot D_1 \ [km], \tag{26}$$

where $EC_{TtWb}$ is the relevant emission coefficient, W is the weight of the load and $D_1$ is the transport distance.

$E_{1TtWf}$ are calculated as follows:

$$E1TtWf \ [kgCO2e] = ECTtWf \ [kgCO2e/tkm] \cdot W \ [t] \cdot D1 \ [km], \tag{27}$$

where $EC_{TtWf}$ is the relevant emission coefficient, W is the weight of the load and $D_1$ is the transport distance.

The total carbon dioxide emissions produced by one-way transportation are calculated as:

$$E = E_1 + E_2 \ [kg_{CO2e}], \tag{28}$$

where E represents the total carbon dioxide emissions produced by transportation, $E_1$ are the total carbon dioxide emissions produced by the one-way transportation and $E_2$ are the additional carbon dioxide emissions (the penalty for an unloaded vehicle).

$E_2$ are calculated using the WtW approach:

$$E_2 = E_{2WtT} + E_{2TtW} \ [kg_{CO2e}], \tag{29}$$

where $E_{2WtT}$ are the carbon dioxide emissions calculated using the WtT approach and $E_{2TtW}$ are the carbon dioxide emissions calculated using the TtW approach.

$E_{2WtT}$ are the carbon dioxide emissions calculated using the WtT approach and consisting of two components:

$$E_{2WtT} = E_{2WtTb} + E_{2WtTf} \ [kg_{CO2e}], \tag{30}$$

where $E_{2WtTb}$ are the carbon dioxide emissions of biogenic origin calculated using the WtT approach and $E_{2WtTf}$ are the carbon dioxide emissions of fossil origin calculated using the WtT approach.

$E_{2TtW}$ are the carbon dioxide emissions calculated using the TtW approach and consisting of two components:

$$E_{2TtW} = E_{2TtWb} + E_{2TtWf} \ [kg_{CO2e}], \tag{31}$$

where $E_{2TtWb}$ are the carbon dioxide emissions of biogenic origin calculated using the TtW approach and $E_{2TtWf}$ are the carbon dioxide emissions of fossil origin calculated using the TtW approach.

The additional distance $D_2$ as a penalty for an unloaded vehicle is calculated as follows:

$$D_2 \ [km] = D_1 \ [km] \cdot c_d \ [-], \tag{32}$$

where $D_1$ is the transport distance and $c_d$ is the internal coefficient defined by the company.

$E_{2WtTb}$ are calculated as follows:

$$E_{2WtTb} \ [kg_{CO2e}] = EC_{0WtTb} \ [kg_{CO2e}/km] \cdot D_2 \ [km], \tag{33}$$

where $EC_{0WtTb}$ is the relevant emission coefficient for the zero load and $D_2$ is the additional transport distance.

$E_{2WtTf}$ are calculated as follows:

$$E_{2WtTf} \ [kg_{CO2e}] = EC_{0WtTf} \ [kg_{CO2e}/km] \cdot D_2 \ [km], \tag{34}$$

where $EC_{0WtTf}$ is the relevant emission coefficient for the zero load and $D_2$ is the additional transport distance.

$E_{2TtWb}$ are calculated as follows:

$$E_{2TtWb} \ [kg_{CO2e}] = EC_{0TtWb} \ [kg_{CO2e}/km] \cdot D_2 \ [km], \tag{35}$$

where $EC_{0TtWb}$ is the relevant emission coefficient for the zero load and $D_2$ is the additional transport distance.

$E_{2TtWf}$ are calculated as follows:

$$E_{2TtWf} \ [kg_{CO2e}] = EC_{0TtWf} \ [kg_{CO2e}/km] \cdot D_2 \ [km], \tag{36}$$

where $EC_{0TtWf}$ is the relevant emission coefficient for the zero load and $D_2$ is the additional transport distance.

*3.6. Framework for Sulfur Dioxide Emissions Calculations in the Context of Road Freight Transport for the Automotive Industry Proposal*

Sulfur dioxide emissions are calculated for a specific shipment. In the case of round trip transportation, Equations (37)–(44) are used. In the case of one-way transportation, Equations (45)–(53) are used.

The total sulfur dioxide emissions produced by round trip transportation are calculated as:

$$E = E_1 \ [kg_{SO2e}], \tag{37}$$

where E represents the total sulfur dioxide emissions produced by transportation and $E_1$ are the total sulfur dioxide emissions produced by the round trip transportation.

$E_1$ are calculated using the WtW approach:

$$E_1 = E_{1WtT} + E_{1TtW} \ [kg_{SO2e}], \tag{38}$$

where $E_{1WtT}$ are the sulfur dioxide emissions calculated using the WtT approach and $E_{1TtW}$ are the sulfur dioxide emissions calculated using the TtW approach.

$E_{1WtT}$ are the sulfur dioxide emissions calculated using the WtT approach and consisting of two components:

$$E_{1WtT} = E_{1WtTb} + E_{1WtTf} \ [kg_{SO2e}], \tag{39}$$

where $E_{1WtTb}$ are the sulfur dioxide emissions of biogenic origin calculated using the WtT approach and $E_{1WtTf}$ are the sulfur dioxide emissions of fossil origin calculated using the WtT approach.

$E_{1TtW}$ are the sulfur dioxide emissions calculated using the TtW approach and consisting of two components:

$$E_{1TtW} = E_{1TtWb} + E_{1TtWf} \ [kg_{SO2e}], \tag{40}$$

where $E_{1TtWb}$ are the sulfur dioxide emissions of biogenic origin calculated using the TtW approach and $E_{1TtWf}$ are the sulfur dioxide emissions of fossil origin calculated using the TtW approach.

$E_{1WtTb}$ are calculated as follows:

$$E_{1WtTb} \ [kg_{SO2e}] = EC_{WtTb} \ [kg_{SO2e}/tkm] \cdot W \ [t] \cdot D_1 \ [km], \tag{41}$$

where $EC_{WtTb}$ is the relevant emission coefficient, W is the weight of the load and $D_1$ is the transport distance.

$E_{1WtTf}$ are calculated as follows:

$$E_{1WtTf} \ [kg_{SO2e}] = EC_{WtTf} \ [kg_{SO2e}/tkm] \cdot W \ [t] \cdot D_1 \ [km], \tag{42}$$

where $EC_{WtTf}$ is the relevant emission coefficient, W is the weight of the load and $D_1$ is the transport distance.

$E_{1TtWb}$ are calculated as follows:

$$E_{1TtWb} \ [kg_{SO2e}] = EC_{TtWb} \ [kg_{SO2e}/tkm] \cdot W \ [t] \cdot D_1 \ [km], \tag{43}$$

where $EC_{TtWb}$ is the relevant emission coefficient, W is the weight of the load and $D_1$ is the transport distance.

$E_{1TtWf}$ are calculated as follows:

$$E_{1TtWf} \ [kg_{SO2e}] = EC_{TtWf} \ [kg_{SO2e}/tkm] \cdot W \ [t] \cdot D_1 \ [km], \tag{44}$$

where $EC_{TtWf}$ is the relevant emission coefficient, W is the weight of the load and $D_1$ is the transport distance.

The total sulfur dioxide emissions produced by one-way transportation are calculated as:

$$E = E_1 + E_2 \ [kg_{SO2e}], \tag{45}$$

where E represents the total sulfur dioxide emissions produced by transportation, $E_1$ are the total sulfur dioxide emissions produced by the one-way transportation and $E_2$ are the additional sulfur dioxide emissions (the penalty for an unloaded vehicle).

$E_2$ are calculated using the WtW approach:

$$E_2 = E_{2WtT} + E_{2TtW} \; [kg_{SO2e}], \tag{46}$$

where $E_{2WtT}$ are the sulfur dioxide emissions calculated using the WtT approach and $E_{2TtW}$ are the sulfur dioxide emissions calculated using the TtW approach.

$E_{2WtT}$ are the sulfur dioxide emissions calculated using the WtT approach and consisting of two components:

$$E_{2WtT} = E_{2WtTb} + E_{2WtTf} \; [kg_{SO2e}], \tag{47}$$

where $E_{2WtTb}$ are the sulfur dioxide emissions of biogenic origin calculated using the WtT approach and $E_{2WtTf}$ are the sulfur dioxide emissions of fossil origin calculated using the WtT approach.

$E_{2TtW}$ are the sulfur dioxide emissions calculated using the TtW approach and consisting of two components:

$$E_{2TtW} = E_{2TtWb} + E_{2TtWf} \; [kg_{SO2e}], \tag{48}$$

where $E_{2TtWb}$ are the sulfur dioxide emissions of biogenic origin calculated using the TtW approach and $E_{2TtWf}$ are the sulfur dioxide emissions of fossil origin calculated using the TtW approach.

The additional distance $D_2$ as a penalty for an unloaded vehicle is calculated as follows:

$$D_2 \; [km] = D_1 \; [km] \cdot c_d \; [-], \tag{49}$$

where $D_1$ is the transport distance and $c_d$ is the internal coefficient defined by the company.

$E_{2WtTb}$ are calculated as follows:

$$E_{2WtTb} \; [kg_{SO2e}] = EC_{0WtTb} \; [kg_{SO2e}/km] \cdot D_2 \; [km], \tag{50}$$

where $EC_{0WtTb}$ is the relevant emission coefficient for the zero load and $D_2$ is the additional transport distance.

$E_{2WtTf}$ are calculated as follows:

$$E_{2WtTf} \; [kg_{SO2e}] = EC_{0WtTf} \; [kg_{SO2e}/km] \cdot D_2 \; [km], \tag{51}$$

where $EC_{0WtTf}$ is the relevant emission coefficient for the zero load and $D_2$ is the additional transport distance.

$E_{2TtWb}$ are calculated as follows:

$$E_{2TtWb} \; [kg_{SO2e}] = EC_{0TtWb} \; [kg_{SO2e}/km] \cdot D_2 \; [km], \tag{52}$$

where $EC_{0TtWb}$ is the relevant emission coefficient for the zero load and $D_2$ is the additional transport distance.

$E_{2TtWf}$ are calculated as follows:

$$E_{2TtWf} \; [kg_{SO2e}] = EC_{0TtWf} \; [kg_{SO2e}/km] \cdot D_2 \; [km], \tag{53}$$

where $EC_{0TtWf}$ is the relevant emission coefficient for the zero load and $D_2$ is the additional transport distance.

### 3.7. Framework for Greenhouse Gas Emissions Calculations in the Context of Road Freight Transport of the Material for the Automotive Industry Application

The application of the framework for GHG emissions calculations in the context of road freight transport of the material for the automotive industry is presented in the form of the interpretative case study in a leading company in the automotive industry operating on the market in the Czech Republic.

Produced GHG emissions are, for example, calculated for the road freight transport with the following parameters:

- Transportation of the material;
- Vehicle $V_4$ ($W_{max}$ = 24,000 kg, $LV_{max}$ = 150 m$^3$);
- Emission standard—EURO 6;
- Weight of the material W = 21,120 kg;
- Volume of the material LV = 125 m$^3$;
- Transport distance $D_1$ = 275 km;
- Type of transport—round trip transport.

Concerning the transported weight of the material, the following restrictive condition and limitation must be respected (according to Equation (1)):

$$\text{for } V_4: 21,120 \leq W_{max} \leq 24,000 \text{ [kg]}, \tag{54}$$

where $V_4$ is the vehicle for the transportation of material and $W_{max}$ is the maximum load weight of the vehicle. The restrictive condition and limitation are met.

Concerning the transported volume of the material, the following restrictive condition and limitation must be respected (according to Equation (2)):

$$\text{for } V_4: 125 \leq LV_{max} \leq 150 \text{ [m}^3\text{]}, \tag{55}$$

where $V_4$ is the vehicle for the transportation of material and $LV_{max}$ is the maximum load volume of the vehicle. The restrictive condition and limitation are met.

The vehicle load factor is calculated (according to Equation (3)) as follows:

$$LF_4 = \text{round up } [21,120/24,000] = 0.88 \text{ [--]}, LF_4 \in <0;1>, \tag{56}$$

where $LF_4$ is the vehicle load factor of vehicle $V_4$.

The emission coefficients are searched in the database according to the following algorithmic procedure (Equation (6)):

$$\begin{array}{c}\text{IF vehicle } V_4 \text{ to } V_7 \text{ AND emission}\\ \text{standard "EURO 6" THEN emission coefficients for "EURO 6" 24 t vehicles,}\end{array} \tag{57}$$

where $V_{4–7}$ represent vehicles for the transportation of the material.

The emission coefficients are further searched in the database according to the following algorithmic procedures (Equation (9)) concerning the type of transport:

$$\begin{array}{c}\text{IF round trip transport THEN find values of emission coefficients}\\ \text{for } EC_{WtTb}, EC_{WtTf}, EC_{TtWb} \text{ and } EC_{TtWf},\end{array} \tag{58}$$

where $EC_{WtTb}$ is the relevant emission coefficient of the biogenic origin calculated using the WtT approach, $EC_{WtTf}$ is the relevant emission coefficient of the fossil origin calculated using the WtT approach, $EC_{TtWb}$ is the relevant emission coefficient of the biogenic origin calculated using the TtW approach, and $EC_{TtWf}$ is the relevant emission coefficient of the fossil origin calculated using the TtW approach.

Emission coefficients were searched for $LF_4$ = 0.88 in the database [57] of carbon dioxide emission coefficients (based on Equations (56)–(58)):

- $EC_{WtTb}$ = 0.00033206 [kg$_{CO2e}$/tkm];

- $EC_{WtTf} = 0.00741130$ [$kg_{CO2e}$/tkm];
- $EC_{TtWb} = 0.00310000$ [$kg_{CO2e}$/tkm];
- $EC_{TtWf} = 0.04240000$ [$kg_{CO2e}$/tkm].

Emission coefficients were searched for $LF_4 = 0.88$ in the database [57] of sulfur dioxide emission coefficients (based on Equations (56)–(58)):

- $EC_{WtT} = 0.00003268$ [$kg_{SO2e}$/tkm];
- $EC_{TtW} = 0.00000996$ [$kg_{SO2e}$/tkm].

Carbon dioxide emissions are calculated for a specific transportation with defined parameters. In this case, it is round trip transportation and Equations (20)–(27) are used.

$E_{1WtTb}$ are calculated as follows (based on Equation (24)):

$$E_{1WtTb} = 0.00033206 \text{ [kg}_{CO2e}/\text{tkm]} \cdot 21.12 \text{ [t]} \cdot 275 \text{ [km]} = 1.92860448 \text{ [kg}_{CO2e}]. \tag{59}$$

$E_{1WtTf}$ are calculated as follows (based on Equation (25)):

$$E_{1WtTf} = 0.00741130 \text{ [kg}_{CO2e}/\text{tkm]} \cdot 21.12 \text{ [t]} \cdot 275 \text{ [km]} = 43.04483040 \text{ [kg}_{CO2e}]. \tag{60}$$

$E_{1TtWb}$ are calculated as follows (based on Equation (26)):

$$E_{1TtWb} = 0.00310000 \text{ [kg}_{CO2e}/\text{tkm]} \cdot 21.12 \text{ [t]} \cdot 275 \text{ [km]} = 18.00480000 \text{ [kg}_{CO2e}]. \tag{61}$$

$E_{1TtWf}$ are calculated as follows (based on Equation (27)):

$$E_{1TtWf} = 0.04240000 \text{ [kg}_{CO2e}/\text{tkm]} \cdot 21.12 \text{ [t]} \cdot 275 \text{ [km]} = 246.25920000 \text{ [kg}_{CO2e}]. \tag{62}$$

$E_{1WtT}$ are the carbon dioxide emissions calculated using the WtT approach and consisting of two components (based on Equation (22)):

$$E_{1WtT} = 1.92860448 + 43.04483040 = 44.97343488 \text{ [kg}_{CO2e}]. \tag{63}$$

$E_{1TtW}$ are the carbon dioxide emissions calculated using the TtW approach and consisting of two components (based on Equation (23)):

$$E_{1TtW} = 18.00480000 + 246.25920000 = 264.26400000 \text{ [kg}_{CO2e}]. \tag{64}$$

$E_1$ are calculated using the WtW approach (based on Equation (21)):

$$E_1 = 44.97343488 + 264.26400000 = 309.23743488 \text{ [kg}_{CO2e}]. \tag{65}$$

The total carbon dioxide emissions produced by round trip transportation are calculated as (based on Equation (20)):

$$E = 309.23743488 \text{ [kg}_{CO2e}], \tag{66}$$

where E represents the total carbon dioxide emissions produced by transportation.

Sulfur dioxide emissions are calculated for a specific transportation with defined parameters. In this case, it is round trip transportation and Equations (37)–(44) are used.

$E_{1WtT}$ are calculated as follows (based on aggregated Equations (39), (41) and (42)):

$$E_{1WtT} = 0.00003268 \text{ [kg}_{SO2e}/\text{tkm]} \cdot 21.12 \text{ [t]} \cdot 275 \text{ [km]} = 0.18980544 \text{ [kg}_{SO2e}]. \tag{67}$$

$E_{1TtW}$ are calculated as follows (based on aggregated Equations (40), (43) and (44)):

$$E_{1TtW} = 0.00000996 \text{ [kg}_{SO2e}/\text{tkm]} \cdot 21.12 \text{ [t]} \cdot 275 \text{ [km]} = 0.05784768 \text{ [kg}_{SO2e}]. \tag{68}$$

$E_1$ are calculated using the WtW approach (based on Equation (38)):

$$E_1 = 0.18980544 + 0.05784768 = 0.24765312 \ [kg_{SO2e}]. \tag{69}$$

The total sulfur dioxide emissions produced by round trip transportation are calculated as (based on Equation (37)):

$$E = 0.24765312 \ [kg_{SO2e}], \tag{70}$$

where E represents the total sulfur dioxide emissions produced by the transportation.

The results are presented in accordance with the requirements of the company in the following form:

- Total carbon dioxide emissions = 309.23743488 $[kg_{CO2e}]$;
- Average carbon dioxide emissions per 1 km = 1.1244997632 $[kg_{CO2e}/km]$;
- Average carbon dioxide emissions per 1 t = 14.641924 $[kg_{CO2e}/t]$;
- Average carbon dioxide emissions per 1 tonne-kilometer = 0.05324336 $[kg_{CO2e}/tkm]$;
- Total sulfur dioxide emissions = 0.24765312 $[kg_{SO2e}]$;
- Average sulfur dioxide emissions per 1 km = 0.0009005568 $[kg_{SO2e}/km]$;
- Average sulfur dioxide emissions per 1 t = 0.011726 $[kg_{SO2e}/t]$;
- Average sulfur dioxide emissions per 1 tonne-kilometer = 0.00004264 $[kg_{SO2e}/tkm]$.

The proposed framework for GHG emissions calculations in the context of road freight transport of the material for the automotive industry (Section 3.3) has been applied and verified.

*3.8. Framework for Greenhouse Gas Emissions Calculations in the Context of Road Freight Transport of Finished Manufactured Passenger Cars for the Automotive Industry Application*

The application of the framework for GHG emissions calculations in the context of road freight transport of FMPC for the automotive industry is presented in the form of the interpretative case study in a leading company in the automotive industry operating on the market in the Czech Republic.

Produced GHG emissions are, for example, calculated for road freight transport with the following parameters:

- Transportation of FMPC;
- Number of FMPC ($PC_1 = 4$, $PC_2 = 2$, $PC_3 = 1$, $PC_4 = 5$, $PC_5 = 4$, $PC_6 = 2$, $\sum PC_j = 18$);
- Vehicle $V_1$ ($W_{max} = 20{,}000$ kg, $LV_{max} = 8$ or 9 FMPC by type);
- Emission standard—EURO 6;
- Weight of the FMPC W = 31,200 kg;
- Transport distance $D_1 = 530$ km;
- Type of transport—round trip transport.

Concerning the transported weight of FMPC, the following restrictive condition and limitation must be respected (according to Equation (10)):

$$\text{for } V_1: W \leq W_{max} \leq 20{,}000 \ [kg], \tag{71}$$

where $V_1$ is the vehicle for the transportation of FMPC and $W_{max}$ is the maximum load weight of the vehicle.

Concerning the transported volume of FMPC, the following restrictive condition and limitation (according to Equation (11)) must be respected:

$$\text{IF } V_1 \text{ AND } (4 + 2) \geq 3 \text{ THEN for } V_1: LV \leq LV_{max} \leq 9 \ [-], \tag{72}$$

where $V_1$ is the vehicle for the transportation of FMPC, LV is the load volume of the FMPC and $LV_{max}$ is the maximum load volume of the vehicle.

The required number of vehicles LKW for FMPC transport is calculated as follows (according to Equation (12)):

$$\text{IF}_1 \ V_1 \ \text{AND} \ (PC_1 + PC_2) \geq 3 \ \text{THEN round down} \ [(PC_1 + PC_2)/3] = LKW_9, \ LKW_9 \cdot 9 = PLKW_9 \ [-],$$
$$\text{IF}_1 \ V_1 \ \text{AND} \ (4 + 2) \geq 3 \ \text{THEN round down} \ [(4 + 2)/3] = 2,$$
$$2 \cdot 9 = 18 \ [-], \tag{73}$$
$$\text{IF}_2 \ PLKW_9 = \sum PC_j \ \text{THEN} \ LKW_9 = LKW, \ LKW \in N,$$
$$\text{IF}_2 \ 18 = 18 \ \text{THEN} \ 2 = 2, \ LKW \in N,$$

where $V_1$ is the vehicle for the transportation of FMPC, $PC_j$ is the number of FMPC type j, j is the type of transported FMPC, LKW is the number of vehicles for FMPC transport, $LKW_9$ is the number of vehicles for nine FMPC transport and $PLKW_9$ is the number of places in the $LKW_9$ vehicle for FMPC transport. Due to the fact that two vehicles $V_1$ will be used for transport, the restrictive condition and limitation are met (according to Equations (72) and (73)):

$$\text{for} \ V_1: \ 31{,}200 \leq W_{max} \leq (2 \cdot 20{,}000) \ [kg],$$
$$\text{IF} \ V_1 \ \text{AND} \ (4 + 2) \geq 3 \ \text{THEN for} \ V_1: \ 18 \leq LV_{max} \leq (2 \cdot 9) \ [-], \tag{74}$$

where $V_1$ is the vehicle for the transportation of FMPC, $W_{max}$ is the maximum load weight of the vehicle and $LV_{max}$ is the maximum load volume of the vehicle.

The average vehicle load factor is calculated (according to Equation (13)) as follows:

$$LF_1 = \text{round up} \ [31{,}200/40{,}000] = 0.78 \ [-], \ LF_1 \in <0;1>, \ i = <1; \infty), \tag{75}$$

where $LF_1$ is the average vehicle load factor of vehicle $V_1$.

The emission coefficients are searched in the database according to the following algorithmic procedure (Equation (16)):

$$\text{IF vehicle} \ V_1 \ \text{AND emission standard "EURO 6" THEN emission}$$
$$\text{coefficients for "EURO 6" 20 t vehicles,} \tag{76}$$

where $V_1$ is the vehicle for the transportation of FMPC.

The emission coefficients are further searched in the database according to the following algorithmic procedures (Equation (19)) concerning the type of transport:

$$\text{IF round trip transport THEN find values of emission coefficients}$$
$$\text{for} \ EC_{WtTb}, \ EC_{WtTf}, \ EC_{TtWb} \ \text{and} \ EC_{TtWf}, \tag{77}$$

where $EC_{WtTb}$ is the relevant emission coefficient of the biogenic origin calculated using the WtT approach, $EC_{WtTf}$ is the relevant emission coefficient of the fossil origin calculated using the WtT approach, $EC_{TtWb}$ is the relevant emission coefficient of the biogenic origin calculated using the TtW approach, and $EC_{TtWf}$ is the relevant emission coefficient of the fossil origin calculated using the TtW approach.

Emission coefficients were searched for $LF_1 = 0.78$ in the database [57] of carbon dioxide emission coefficients (based on Equations (75)–(77)):

- $EC_{WtTb} = 0.00034244 \ [kg_{CO2e}/tkm]$;
- $EC_{WtTf} = 0.00802115 \ [kg_{CO2e}/tkm]$;
- $EC_{TtWb} = 0.00340000 \ [kg_{CO2e}/tkm]$;
- $EC_{TtWf} = 0.04580000 \ [kg_{CO2e}/tkm]$.

Emission coefficients were searched for $LF_1 = 0.78$ in the database [57] of sulfur dioxide emission coefficients (based on Equations (75)–(77)):

- $EC_{WtT} = 0.00003540 \ [kg_{SO2e}/tkm]$;
- $EC_{TtW} = 0.00001160 \ [kg_{SO2e}/tkm]$.

Carbon dioxide emissions are calculated for a specific transportation with defined parameters. In this case, it is round trip transportation and Equations (20)–(27) are used.

$E_{1WtTb}$ are calculated as follows (based on Equation (24)):

$$E_{1WtTb} = 0.00034244 \ [kg_{CO2e}/tkm] \cdot 31.2 \ [t] \cdot 530 \ [km] = 5.66258784 \ [kg_{CO2e}]. \tag{78}$$

$E_{1WtTf}$ are calculated as follows (based on Equation (25)):

$$E_{1WtTf} = 0.00802115 \ [kg_{CO2e}/tkm] \cdot 31.2 \ [t] \cdot 530 \ [km] = 132.63773640 \ [kg_{CO2e}]. \tag{79}$$

$E_{1TtWb}$ are calculated as follows (based on Equation (26)):

$$E_{1TtWb} = 0.00340000 \ [kg_{CO2e}/tkm] \cdot 31.2 \ [t] \cdot 530 \ [km] = 56.22240000 \ [kg_{CO2e}]. \tag{80}$$

$E_{1TtWf}$ are calculated as follows (based on Equation (27)):

$$E_{1TtWf} = 0.04580000 \ [kg_{CO2e}/tkm] \cdot 31.2 \ [t] \cdot 530 \ [km] = 757.34880000 \ [kg_{CO2e}]. \tag{81}$$

$E_{1WtT}$ are the carbon dioxide emissions calculated using the WtT approach and consisting of two components (based on Equation (22)):

$$E_{1WtT} = 5.66258784 + 132.63773640 = 138.30032424 \ [kg_{CO2e}]. \tag{82}$$

$E_{1TtW}$ are the carbon dioxide emissions calculated using the TtW approach and consisting of two components (based on Equation (23)):

$$E_{1TtW} = 56.22240000 + 757.34880000 = 813.57120000 \ [kg_{CO2e}]. \tag{83}$$

$E_1$ are calculated using the WtW approach (based on Equation (21)):

$$E_1 = 138.30032424 + 813.57120000 = 951.87152424 \ [kg_{CO2e}]. \tag{84}$$

The total carbon dioxide emissions produced by round trip transportation are calculated as (based on Equation (20)):

$$E = 951.87152424 \ [kg_{CO2e}], \tag{85}$$

where E represents the total carbon dioxide emissions produced by transportation.

Sulfur dioxide emissions are calculated for a specific transportation with defined parameters. In this case, it is round trip transportation and Equations (37)–(44) are used.

$E_{1WtT}$ are calculated as follows (based on aggregated Equations (39), (41) and (42)):

$$E_{1WtT} = 0.00003540 \ [kg_{SO2e}/tkm] \cdot 31.2 \ [t] \cdot 530 \ [km] = 0.58537440 \ [kg_{SO2e}]. \tag{86}$$

$E_{1TtW}$ are calculated as follows (based on aggregated Equations (40), (43) and (44)):

$$E_{1TtW} = 0.00001160 \ [kg_{SO2e}/tkm] \cdot 31.2 \ [t] \cdot 530 \ [km] = 0.19181760 \ [kg_{SO2e}]. \tag{87}$$

$E_1$ are calculated using the WtW approach (based on Equation (38)):

$$E_1 = 0.58537440 + 0.19181760 = 0.77719200 \ [kg_{SO2e}]. \tag{88}$$

The total sulfur dioxide emissions produced by round trip transportation are calculated as (based on Equation (37)):

$$E = 0.77719200 \ [kg_{SO2e}], \tag{89}$$

where E represents the total sulfur dioxide emissions produced by transportation.

The results are presented in accordance with the requirements of the company in the following form:

- Total carbon dioxide emissions = 951.87152424 [$kg_{CO2e}$];
- Average carbon dioxide emissions per 1 km = 1.795984008 [$kg_{CO2e}$/km];
- Average carbon dioxide emissions per 1 t = 30.5087027 [$kg_{CO2e}$/t];
- Average carbon dioxide emissions per 1 tonne-kilometer = 0.05756359 [$kg_{CO2e}$/tkm];
- Total sulfur dioxide emissions = 0.77719200 [$kg_{SO2e}$];
- Average sulfur dioxide emissions per 1 km = 0.0014664 [$kg_{SO2e}$/km];
- Average sulfur dioxide emissions per 1 t = 0.02491000 [$kg_{SO2e}$/t];
- Average sulfur dioxide emissions per 1 tonne-kilometer = 0.00004700 [$kg_{SO2e}$/tkm].

The proposed framework for GHG emissions calculations in the context of road freight transport of FMPC for the automotive industry (Section 3.4) has been applied and verified.

## 4. Discussion

The issue of global warming is an increasingly discussed topic due to various human activities. The transport sector is ambivalent because it is indispensable for the state's economy, but on the other hand has negative environmental impacts (emissions, noise, vibration, land use). The transport sector is the second most important sector contributing to the production of $CO_2$ emissions worldwide. Globally, the largest producer of emissions is road transport in terms of transport modes, and emissions from road freight transport continue to rise.

The aim of this article was to propose a framework for greenhouse gas emissions calculations in the context of road freight transport for the automotive industry. The automotive industry has long been the most important sector of the economy in the Czech Republic, but also in the European Union. A global and extensive supply chain with many links is characteristic for the automotive industry. The automotive industry is completely dependent on a large number of suppliers and on transport. Most transports are carried out using road freight transport. This also implies the production of GHG emissions. However, companies in the automotive industry are trying to reduce the negative impact on the environment through logistic processes. Companies take into account the production of GHG emissions in logistic planning and decision-making, but there are no suitable tools for this. Calculators of GHG emissions are important tools for estimating GHG emissions. However, current calculators of GHG emissions do not cover all the transport restrictive conditions and limitations used in the automotive industry. This statement was also confirmed by these results: identification of requirements for the calculation of GHG emissions from road freight transport (Section 3.1), analysis of available emission freight calculators (Section 3.2), and semi-structured interviews (Appendices A and B). A market gap was identified.

The frameworks were proposed based on the results of semi-structured interviews, content analysis and comparative analysis. The framework for GHG emissions calculations in the context of road freight transport of material for the automotive industry was proposed (Section 3.3). The framework for GHG emissions calculations in the context of road freight transport of FMPC for the automotive industry was proposed (Section 3.4). The framework for carbon dioxide emissions calculations in the context of road freight transport for the automotive industry was proposed (Section 3.5). The framework for sulfur dioxide emissions calculations in the context of road freight transport for the automotive industry was proposed (Section 3.6).

The proposed framework for GHG emissions calculations in the context of road freight transport of material for the automotive industry has been applied and verified (Section 3.7). The proposed framework for GHG emissions calculations in the context of road freight transport of FMPC for the automotive industry has been applied and verified (Section 3.8).

The verification of the proposed frameworks was performed using real transportation of FMPC and material. GHG emissions were calculated for the transport of FMPC with the following parameters: vehicle $V_1$, emission standard EURO 6, weight of the cargo 31,200 kg, transport distance 530 km, round trip transport. Total GHG emissions were

calculated for this transportation as follows: 951.871 kgCO$_2$e and 0.777 kgSO$_2$e (Table 3). GHG emissions were calculated for the transport of material with the following parameters: vehicle V$_4$, emission standard EURO 6, weight of the cargo 21,120 kg, transport distance 275 km, round trip transport. Total GHG emissions were calculated for this transportation as follows: 309.237 kgCO$_2$e and 0.247 kgSO$_2$e (Table 3).

**Table 3.** Summary of results [authors].

| Type of Cargo | Vehicle | Emission Standard | Weight [kg] | Distance [km] | Total CO$_2$ Emissions [kg$_{CO2e}$] | Total SO$_2$ Emissions [kg$_{SO2e}$] |
|---|---|---|---|---|---|---|
| FMPC | V$_1$ | EURO 6 | 31,200 | 530 | 951.871 | 0.777 |
| Material | V$_4$ | EURO 6 | 21,120 | 275 | 309.237 | 0.247 |

The proposed frameworks can be used to implement a GHG emissions calculator suitable for the automotive industry because the proposed frameworks include all the road freight transport restrictive conditions and limitations used in the automotive industry, unlike other available calculators. Automotive industry companies could better take into account the production of GHG emissions in logistic planning and decision-making.

This approach is necessary because there is no approach or available emission calculator suitable for GHG emissions calculations from road freight transport customized for the automotive industry. This was also confirmed by the results of semi-structured interviews and the results of the analysis of available emission freight calculators. The most important conclusions were as follows: the company does not currently use any road transport emissions freight calculators because there is no suitable solution available to meet the company's requirements. The logistic processes of the company are very extensive and specific with many conditions. Currently, there is no freight emission calculator that contains all specifics. This approach and the proposed frameworks fully respect the three main approaches to measuring energy consumption and produced emissions—WtW, WtT and TtW.

There can be some possible limitations in our approach. The processed theoretical backgrounds do not cover all available sources in connection with the given issue, but significant sources in the given area were selected using the method of literary research.

Another potential limitation of the processed approach is the identification of requirements for the calculation of GHG emissions from road freight transport in the automotive industry. These requirements were identified by two independent researchers using the two-round semi-structured interview with three respondents from a leading company in the automotive industry operating on the market in the Czech Republic. On the one hand, the limitation may lie in the number of respondents, and on the other hand, the limit may be the use of all respondents from one company. However, this approach is relevant because important experts in the field were chosen as respondents. At the same time, the respondents were selected from the leading company in the automotive industry operating on the market in the Czech Republic, where the interpretative case study was also processed. This approach is also relevant due to the fact that automotive industry companies have the same or very similar inbound, in-house and outbound processes.

Another limiting fact may be the selection of emission freight calculators for analysis because not all existing emission freight calculators were affected. It must be stated that the selection of emission freight calculators was made objectively using Google search by three independent researchers. On the other hand, there could be other emission freight calculators that have not been analyzed.

The limitation of this article can also lie in the proposed frameworks (Framework for greenhouse gas emissions calculations in the context of road freight transport of the material and FMPC for the automotive industry; Framework for carbon dioxide and sulfur dioxide emissions calculations in the context of road freight transport for the automotive industry proposal). The idea of proposed frameworks is based on the literature background and

the results are influenced by the results of semi-structured interviews for the identification of requirements for the calculation of GHG emissions from road freight transport in the automotive industry. These frameworks can therefore be used to calculate GHG emissions from road freight transport in the automotive industry, but they can also be appropriated for other companies in the sector to customize these frameworks.

## 5. Conclusions

Today, it is quite clear that the issue of global warming is a key issue for this millennium. Thanks to this, it is also necessary to pay attention to the GHG emissions and human activities that cause global warming. Transport, as a key sector of the national economy, causes economic growth but also negative environmental effects. Thanks to this, it is necessary to pay great attention to transport, especially road freight transport, in terms of the volume of GHG emissions produced.

The issue of greenhouse gas emissions is also related to the automotive industry. This industry is characterized by an extensive and global supply chain. This fact also brings with it a large number of transports, usually carried out by road freight transport. Automotive industry companies, on the other hand, strive to behave sustainably and responsibly and to take GHG emissions into account in logistic decisions and planning. Given that the issue of calculating GHG emissions from transport is constantly evolving, in the future it will be necessary to look for new tools to support logistic planning and decision-making. This area of scientific interest is expected to expand further.

The aim of this article was to propose a framework for greenhouse gas emissions calculations in the context of road freight transport for the automotive industry. First, the requirements for the calculation of greenhouse gas emissions from road freight transport were identified; second, available emission freight calculators were analyzed; third, a framework for greenhouse gas emissions calculations in the context of road freight transport of material and finished manufactured passenger cars for the automotive industry was proposed; fourth, a framework for carbon dioxide and sulfur dioxide emissions calculations in the context of road freight transport for the automotive industry was proposed; and fifth, a framework for greenhouse gas emissions calculations in the context of road freight transport of material and finished manufactured passenger cars for the automotive industry was applied.

This research clearly demonstrated the non-existence of an approach or available emission freight calculators suitable for GHG emissions calculations from road freight transport customized for the automotive industry. Furthermore, all significant specifications, parameters, restrictive conditions or limitations related to the road freight transport within the automotive industry were identified. At the same time, frameworks were proposed that respect both the approaches to calculating GHG emissions and all the essential specifics of the automotive industry.

Future research steps will concern the field of rail transport in relation to the produced GHG emissions and their calculations because rail transport is also widely used for transport within the automotive industry, especially for outbound logistic processes. At the same time, there is no customized approach or emission calculator for the GHG emissions calculations in the context of railway freight transport for the automotive industry. Another area of future research is the proposal of customized frameworks for other sectors or industries, which monitor, evaluate and make decisions depending on the volume of GHG emissions produced, e.g., chemical and pharmaceutical industries.

**Author Contributions:** Conceptualization, R.H. and S.M.; methodology, J.C. and D.S.; validation, S.M. and P.S.; formal analysis, D.S.; resources, R.H. and D.S.; writing—original draft preparation, J.C.; writing—review and editing, R.H. and S.M.; visualization, S.M.; project administration, J.C. All authors have read and agreed to the published version of the manuscript.

**Funding:** This research was funded by the project "Cooperation in Applied Research between the University of Pardubice and Companies, in the Field of Positioning, Detection and Simulation Technology for Transport Systems (PosiTrans)", registration No.: CZ.02.1.01/0.0/0.0/17_049/0008394.

**Institutional Review Board Statement:** Not applicable.

**Informed Consent Statement:** Not applicable.

**Data Availability Statement:** Data are contained within the article.

**Acknowledgments:** This article is published within the realization of the project "Cooperation in Applied Research between the University of Pardubice and Companies, in the Field of Positioning, Detection and Simulation Technology for Transport Systems (PosiTrans)", registration No.: CZ.02.1.01/0.0/0.0/17_049/0008394.

**Conflicts of Interest:** The authors declare no conflict of interest.

## Appendix A

Questions used in semi-structured interviews:

1. Is the issue of calculating GHG emissions from road transport relevant for you?
2. Do you use a GHG emissions calculator?
3. In what logistic processes do you monitor the GHG emissions produced by road transport (inbound logistics, in-house logistics and outbound logistics)?
4. What type of cargo do you transport using road freight transport (e.g., products, materials, containers, etc.)?
5. What types of vehicles do you use for transport?
6. What are the specifications and parameters of the vehicles used for transport?
7. What are the other restrictive conditions or limitations for transport?
8. What type of transport do you carry out (one-way transport, round trip (return) transport)?
9. What GHG emissions do you monitor in logistic processes?
10. How do you want to present the resulting GHG emissions (e.g., total emissions, average emissions per 1 km, average emissions per 1 t and average emissions per 1 tonne-kilometer)?
11. Do you use the conversion (monetization) of produced emissions into financial value?

## Appendix B

Aggregated answers obtained from semi-structured interviews:

1. Is the issue of calculating GHG emissions from road transport relevant for you?
2. Yes, this issue is very current and crucial for our company. We strive to be a "green company and factory".
3. Do you use a GHG emissions calculator?
4. We do not currently use any road transport emissions freight calculators, as there is no suitable calculator available to meet our requirements. Our logistic processes are very extensive and specific with many conditions. Currently, there is no freight emission calculator that contains all specifics.
5. In what logistic processes do you monitor the GHG emissions produced by road transport (inbound logistics, in-house logistics and outbound logistics)?
6. We monitor GHG emissions produced by road freight transport in inbound logistic processes, in-house logistic processes and outbound logistic processes.
7. What type of cargo do you transport using road freight transport (e.g., products, materials, containers, etc.)?
8. We transport finished products (FMPC) using road freight transport and many other items that can be included in the material.
9. What types of vehicles do you use for transport?

10. We use two groups of vehicles. The first group of vehicles is used for the transportation of FMPC. The second group of vehicles is used for the transportation of material. An overview of these vehicles is shown in Table 1.
11. What are the specifications and parameters of the vehicles used for transport?
12. An overview of the specifications and parameters of the vehicles used for road freight transport is shown in Table 1.
13. What are the other restrictive conditions or limitations for transport?
14. The maximum load weight of the vehicle, the maximum load volume of the vehicle and vehicle selection by type of cargo must be respected. Specific conditions apply to the transport of FMPC. We transport a total of 8 types of FMPC ($PC_1$–$PC_8$). The following restrictions apply to the FMPC:

    - If the FMPC are transported in a vehicle ($V_1$) and at least 3 cars ($PC_1$–$PC_2$) are loaded at the same time, then it is possible to load a total of 9 FMPC;
    - If the condition above is not met, it is possible to load in a vehicle ($V_1$) a total of 8 FMPC;
    - If the FMPC are transported in a vehicle ($V_2$), then it is possible to load a total of 4 FMPC;
    - If the FMPC are transported in a vehicle ($V_3$), then it is possible to load a total of 2 FMPC.

15. What type of transport do you carry out (one-way transport, round trip (return) transport)? We provide one-way transport and round trip (return) transport. In the case of one-way transport, we multiply the produced emissions by an internal coefficient because it is a penalty for an unloaded vehicle.
16. What GHG emissions do you monitor in logistic processes? We monitor carbon dioxide ($CO_2$) and sulfur dioxide ($SO_2$) emissions as part of logistics processes.
17. How do you want to present the resulting GHG emissions (e.g., total emissions, average emissions per 1 km, average emissions per 1 t and average emissions per 1 tonne-kilometer)? We work with the following emission values: total emissions, average emissions per 1 km, average emissions per 1 t and average emissions per 1 tonne-kilometer. We further distinguish emissions according to the calculation approach (Well-to-Wheel, Well-to-Tank and Tank-to-Wheel approach) and according to the origin of emissions (fossil and biogenic origin).
18. Do you use the conversion (monetization) of produced emissions into financial value? We use the conversion (monetization) of produced emissions into financial value by the internal price of produced emissions as part of internal company calculations.

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
