# Peer review of "Framework for Greenhouse Gas Emissions Calculations in the Context of Road Freight Transport for the Automotive Industry"

_sustainability, doi:10.3390/su13074068_

Round 1

Reviewer 1 Report

The document presents a good quality research. However, some reorganization of the paper would help to understand the procedure and also to improve the presentation of final conclusions. 

Some parts of results Section, would be better allocated in the method Section. Thus, leaving in results what is results, and leaving equations in the method Section. 

Formulation could be better understood if grouped with suffixes, especially in those cases in which equations are the same for several situations, using the typical:  Vi, for i=1...5 (for example). That way, the total number of equations is not as high as it is now. 

Discussion and conclusions could be extended in order to connect the final of the paper with the aim of the paper. 

Author Response

Reviewer 1

Author´s response: “Framework for greenhouse gas emissions calculations in the context of road freight transport for the automotive industry”

Dear and esteemed reviewer,

First, let us thank you for reviewing the article, providing feedback, and sending inspiring comments and suggestions. We tried our best to incorporate all the comments and we send as we incorporated them.

Some parts of results Section, would be better allocated in the method Section. Thus, leaving in results what is results, and leaving equations in the method Section.

We did not respectfully dare to incorporate this comment. We assume that Chapter 2 (Materials and Methods) contains a theoretical description of the materials and methods used to process the article. In addition, the results of using these methods are given below in Chapter 3 (Results). The equations themselves are not a method, but the result of a solution, respectively they are part of the proposed framework. For these reasons, we have left the theoretical description of the methods in Chapter 2 (Materials and Methods) and the results of the use of these methods, including the equations in Chapter 3 (Results).

Formulation could be better understood if grouped with suffixes, especially in those cases in which equations are the same for several situations, using the typical:  Vi, for i=1...5 (for example). That way, the total number of equations is not as high as it is now.

We did not respectfully dare to incorporate this comment. Although the equations look similar at first glance, they are different or account for different variables and cannot be grouped with suffixes together. The equations in the article can be explained as follows for better understanding:

  • restrictive conditions and limitations to the transportation of material (equations 1-2),
  • load factor calculation and algorithmic expression of the calculation process (equations 3-9),
  • restrictive conditions and limitations to the transportation of FMPC (equations 10-12),
  • load factor calculation and algorithmic expression of the calculation process (equations 13-19),
  • framework for carbon dioxide emissions calculations (equations 20-36),
  • framework for sulfur dioxide emissions calculations (equations 37-53),
  • framework to the transportation of material application (equations 54-70),
  • framework to the transportation of FMPC application (equations 71-89).

Discussion and conclusions could be extended in order to connect the final of the paper with the aim of the paper.

The Discussion has been significantly improved, expanded and more elaborated (lines 734-742 and 751-791). At the same time, article limits have been added (lines 765-791). The conclusion has been significantly improved (lines 807-832). The conclusion was expanded and more elaborated. It was further better connected to the aim of the article (lines 807-817) and was supplemented by the results (lines 818-824) and future research steps (lines 825-832) in relation to the content of the article and main findings.

Thank you very much.

Pardubice, 24th March 2020     

Authors

Reviewer 2 Report

The paper is interesting and provides useful content for the bibliography. Suggestions for improving it are:
- cite study Marinello et al., 2020. Roadway tunnels: A critical review of air pollutant concentrations and vehicular emissions. https://www.sciencedirect.com/science/article/pii/S1361920920306659
- Explain better why this approach may be necessary and better than already available approaches (eg https://www.eea.europa.eu/help/glossary/eea-glossary/corinair)
-improve conclusions

Author Response

Reviewer 2

Author´s response: “Framework for greenhouse gas emissions calculations in the context of road freight transport for the automotive industry”

Dear and esteemed reviewer,

First, let us thank you for reviewing the article, providing feedback, and sending inspiring comments and suggestions. We tried our best to incorporate all the comments and we send as we incorporated them.

Cite study Marinello et al., 2020. Roadway tunnels: A critical review of air pollutant concentrations and vehicular emissions. https://www.sciencedirect.com/science/article/pii/S1361920920306659.

The interesting study “Marinello, S.; Lolli, F.; Gamberini, R. Roadway tunnels: A critical review of air pollutant concentrations and vehicular emissions. Transportation Research Part D 2020, 86. https://doi.org/10.1016/j.trd.2020.102478” was cited in the text (line 61) and was added as a reference no. 9 (lines 941-942).

Explain better why this approach may be necessary and better than already available approaches (eg https://www.eea.europa.eu/help/glossary/eea-glossary/corinair).

This explanation is given in the article on the lines 226-232 and 263-267. This explanation has been improved, expanded and elaborated (lines 751-761).

Improve conclusions.

The conclusion has been significantly improved (lines 807-832). The conclusion was expanded and more elaborated. It was further better connected to the aim of the article (lines 807-817) and was supplemented by the results (lines 818-824) and future research steps (lines 825-832) in relation to the content of the article.

Thank you very much.

Pardubice, 24th March 2020

Authors

Reviewer 3 Report

Dear Authors,

Thank you for sending your paper to the journal Sustainability. I find your paper quite interesting and like to propose some improvements:

  1. Line 36 try to avoid multiple citations. It is always good to explain each reference.
  2. Line 143 see the previous comment.
  3. Line 190 – 207 is an entirely unnecessary part. You need at least one paragraph in between, but this is not away.
  4. Regarding 3.2 For sure, the first decision is that you can use a calculator for road transport. So, in this case, maybe it is better to make a table after this step. So, you can rewrite the paragraph and make the table much easier to read.
  5. Page 9 to 13 is hard to read. Maybe a block diagram is a solution with comments.
  6. Results form chapter 3 can be represented in one table in chapter 4.
  7. In Disucssion it is not commen to use citaiotns. This chapter must discuss your methodology, results, pro et contra, …
  8. The conclusion is quite generic. Add some of your results and future research steps according to your findings.

Regards,

Author Response

Reviewer 3

Author´s response: “Framework for greenhouse gas emissions calculations in the context of road freight transport for the automotive industry”

Dear and esteemed reviewer,

First, let us thank you for reviewing the article, providing feedback, and sending inspiring comments and suggestions. We tried our best to incorporate all the comments and we send as we incorporated them.

Ad 1. Line 36 try to avoid multiple citations. It is always good to explain each reference.

Multiple citations (originally on the line 36) were removed and each source is now explained separately (lines 39-41).

Ad 2. Line 143 see the previous comment.

Multiple citations (originally on the line 143) were removed and each source is now explained separately (lines 146-151).

Ad 3. Line 190 – 207 is an entirely unnecessary part. You need at least one paragraph in between, but this is not away.

Lines, originally 190 - 207, have been completely reworked and rewritten (lines 198-207). This section of the article we consider very important because it helps readers orientation in Chapter 3 (Results). Due to this fact, we have greatly simplified the text, but it still describes the structure of Chapter 3 (Results). 

Ad 4. Regarding 3.2 For sure, the first decision is that you can use a calculator for road transport. So, in this case, maybe it is better to make a table after this step. So, you can rewrite the paragraph and make the table much easier to read.

We have completely reworked and simplified subchapter 3.2 (lines 234-251). We have also significantly modified and simplified Table 2 (lines 252-253).

Ad 5. Page 9 to 13 is hard to read. Maybe a block diagram is a solution with comments.

The block diagram (Figure 4, lines 285-286) is a solution - Framework for greenhouse gas emissions calculations in the context of road freight transport of material and FMPC for the automotive industry proposal. The Figure 4 schematically shows the process of greenhouse gas emissions calculations in the context of road freight transport for the automotive industry. We have significantly simplified the text in subchapter 3.3 and outlined the individual steps of the framework (lines 270-285). The next part of the subchapter (lines 289-331) contains an algorithmic expression of the proposed calculation, including limiting conditions.

Ad 6. Results form chapter 3 can be represented in one table in chapter 4.

This comment was also fully incorporated. Table 3 has been added (lines 743-744) to Chapter 4 (Discussion). The text related to this table has also been added to Chapter 4 (Discussion) (lines 734-742).

Ad 7. In Disucssion it is not commen to use citaiotns. This chapter must discuss your methodology, results, pro et contra, …

The citations have been completely removed from Chapter 4 (Discussion) (lines 696-733). The Discussion has been significantly improved, expanded and more elaborated (lines 734-742 and 751-791). At the same time, article limits have been added (lines 765-791).

Ad 8. The conclusion is quite generic. Add some of your results and future research steps according to your findings.

The conclusion has been significantly improved (lines 807-832). The conclusion was expanded and more elaborated. It was further better connected to the aim of the article (lines 807-817) and was supplemented by the results (lines 818-824) and future research steps (lines 825-832) in relation to the content of the article and main findings.

Thank you very much.

Pardubice, 24th March 2020     

Authors

Round 2

Reviewer 3 Report

Dear Authors,

Thank you for the new version of your paper. You have proactively answered all my raised questions.

Regards,